# Bristol girls dance project feasibility study: using a pilot economic evaluation to inform design of a full trial

Jane E Powell,[1] Fran E Carroll,[2] Simon J Sebire,[3] Anne M Haase,[3] Russell Jago[3]

[1]Department of Health and Social Sciences, University of the West of England, Bristol, UK
[2]F E Carroll School of Social and Community Medicine, University of Bristol, Bristol, UK
[3]Centre for Exercise Nutrition and Health Sciences, School for Policy Studies, University of Bristol, Bristol, UK

**Correspondence to**
Professor Jane E Powell;
jane.powell@uwe.ac.uk

## ABSTRACT

**Background:** There is currently little guidance for pilot trial economic evaluation where health outcomes and costs are influenced by a range of wider determinants and factors.

**Objectives:** This article presents the findings of a pilot economic evaluation study running alongside the Bristol Girls Dance Project (BGDP) feasibility study.

**Design:** 3-arm, cluster randomised, controlled pilot trial and economic evaluation. 7 schools (n=210) from the Bristol and greater Bristol area, UK were randomly allocated to the intervention arm 3 schools (n=90) and the control arm 4 schools (n=120).

**Intervention:** Girls aged 11–12 years with parental consent were provided with two, 90 min dance sessions per week for 9 weeks at school facilities.

**Economic outcome measures:** Programme costs and girls' preferences for attributes of dance and preferences for competing leisure time activities were measured.

**Results:** The mainstream average cost of the BGDP programme (not including research, control and dance teacher training costs) per school was $2126.40, £1329 and €1555 and per participant was $70.90, £44.31 and €51.84 in 2010–2011 prices. Discrete choice experiment (DCE) methods are acceptable to girls of this age indicating time available for other leisure activities on dance class days is the attribute girls valued most and 2 h leisure time remaining preferred to 3 h.

**Conclusions:** This pilot study indicates that providing full cost data for a future trial of the BGDP programme is feasible and practical. There is no evidence from preference data to support adjustment to intervention design. A future economic evaluation is likely to be successful utilising the resource use checklist developed. The importance of categorising separately resources used to develop, prepare, deliver and maintain the programme to estimate mainstream costs accurately is demonstrated.

## BACKGROUND

Recent influential studies attach substantial economic and social costs to obesity prevalence projections.[1] [2] These forecasts are based on a body of research from long-term cohort studies which suggest that change in the

### Strengths and limitations of this study

This pilot study used a systematic approach where there is currently minimal evidence to determine the costs of implementing a pilot dance intervention in girls aged 11–12 years. The study has produced findings about girls' preferences for dance and an embryonic costing tool that can be applied to design and conduct an economic evaluation alongside a full cluster randomised controlled trial (RCT). This feasibility and exploratory pilot study is powered to test the intervention concept, the feasibility of obtaining programme cost data in categories and the evidence required to power a full cluster RCT in the future. Consequently, the variation in programme costs at the school level has not been captured and this is a limitation of the cost estimates presented.

prevalence of obesity in children and adolescents born at the millennium is likely to lead to increased health risks in middle age irrespective of adult adiposity.[3–5] Consequently, there is a need for new interventions that focus on preventing obesity or changing diet or physical activity (PA), the two behaviours that are central to the accrual of body mass.

As well as being a health and well-being issue, children's obesity also has serious economic impacts. Scarce resources with competing uses in all health systems and the need to decide between new, 'efficacious' primary prevention PA programme interventions on the grounds of cost-effectiveness have increased the significance of economic evaluation as a concept and methodology. Recent guidance from the UK Medical Research Council (MRC) for the development and evaluation of complex behavioural interventions suggests that efficacy and cost-effectiveness should be established before programmes are implemented at the population level.[6] [7] However, the meaningful determination of these criteria is often problematic in primary prevention, and guidelines for the design and conduct of economic evaluation of complex interventions are at an early stage of development.[8–11] It is,

**1**

therefore, important to develop the conceptual and measurement process by which effectiveness and cost-effectiveness of complex PA interventions can be evaluated in a full trial using a pilot study.

The main findings of the Bristol Girls Dance Project (BGDP) feasibility trial concerning process evaluation, outcomes and effectiveness have been published elsewhere.[12] This part of the study suggested that it is feasible to deliver the intervention and that participating in dance has the potential to yield change in moderate-to-vigorous PA (MVPA) among 11–12-year-old girls (school year 7), but a larger randomised controlled trial (RCT) would be needed to fully evaluate its effectiveness and cost-effectiveness.[12] In the absence of robust evidence for the cost and outcome of dance interventions, other aims of the feasibility pilot were to refine the information required to sufficiently power a full trial and to use the preference data to inform potential refinements to intervention design.

Preferences for competing after-school activities are potential determinants of the economic benefit of dance intervention, as increased PA must be valued in order for it to be maintained[8] and to have potential for long-term impact on PA levels. In this study, discrete choice experiment (DCE) and survey methods are applied to examine two separate, but complementary aspects of value—preferences for the attributes of dance classes and preferences for dance among other competing alternatives for spending leisure time, respectively. PA levels decline during youth[13] with the start of secondary school being a critical period of change, so it was important to establish comparative preferences for after-school leisure activities on weekdays.

Value is a concept germane to recruitment and retention rates and linked to the outcome dimension of the BGDP intervention and is therefore important to examine in detail. DCE works on the premise that any 'product', for example, a healthcare treatment or PA programme, can be described by its characteristics or attributes, and the extent to which an individual values a 'product' is dependent on the level of these characteristics.[14–16]

Thus, this article reports the findings of a pilot economic evaluation of the BGDP for girls aged 11–12 years in a primary school setting in England that can be applied to design and conduct a future full trial and economic evaluation.

Two hours available for other leisure activities on dance class days was preferred to 3 h, suggesting after-school dance classes are valued compared with other ways to spend leisure time after school on weekdays.

Resources used in the development, preparation, delivery and maintenance of dance classes should be categorised separately in stages in order to identify the mainstream cost of the programme intervention to commissioners.

## METHODS
### BGDP feasibility study
BGDP was a three-arm, parallel group, cluster randomised, controlled pilot trial with schools as the unit of allocation. Seven schools from three school districts, Bristol, Bath and South Gloucestershire (the UK), were recruited to take part in the study from schools in these districts with no current after-school dance provision.[12] The hip-hop and street dance style of dance to popular music was facilitated by a professional dance teacher.

Stratifying by school district, three schools were randomly allocated to the intervention arm (n=90) and four schools to the two control arms (n=120) and each school was assigned a dance teacher to lead the sessions. Randomisation was conducted by an independent member of the clinical trials unit at Bristol University using computer-generated random sequences and codes for school district and school name. The three intervention schools received two, 90 min after-school dance classes per week for 9 weeks selected to allow the entire programme to be delivered within a school term. Pilot work had suggested that dance is a very attractive form of PA for girls, so the control element was designed to ascertain whether offering a dance workshop at the end of the research process (ie, after the last data collection) would affect either retention or the quality of data provided by participants. We, therefore, utilised a three-arm design with two different control groups. In two schools, participants were provided with small thank you gifts for each wave of data collection. In the other two control schools, participants were provided with the same small thank you gifts, as well as a half-day dance workshop at the end of the study.

### Sample size
This feasibility study was powered to test the intervention concept and to provide the necessary information to calculate the sample size of a full cluster RCT and economic evaluation of an after-school dance programme. Detection of important parameters of 10 min difference in MVPA per weekday (50 min/week) between the intervention and control groups, intraclass correlation for weekday MVPA at time 2 and associated CIs have been reported and profiled in another article from this study.[12] For practical reasons, the sample was limited to 30 girls aged 11–12 years per school. Girls were recruited from each school at random from those with parental consent.

### Economic measures
#### DCE and survey of preference ranking and use of leisure time
BGDP formative qualitative work indicated the frequency of after-school dance classes per week, cost per session and how much leisure time is left over on dance class days for other leisure activities are important considerations for girls in deciding whether to participate.[17] Participants were asked to select the 'dance class scenario' they preferred from a pair of options. Table 1 presents the four paired scenarios (1A:1B, 2A:2B, 3A:3B and 4A:4B) consisting of a randomly determined combination of three attributes, each with two levels.

**Table 1** Attributes and levels of the discrete choice experiment and the four choice sets given to participants

| Level of attributes | Attributes | | |
| --- | --- | --- | --- |
| | Frequency of dance classes/week | Cost/ session | Hours left for other leisure activities on that day |
| Upper | 2 dance classes/week | £1/session | Leaving 3 h for other leisure activities on the evening of the dance session |
| Lower | 1 dance class/week | 50p/session | Leaving 2 h for other leisure activities on the evening of the dance session |

1A↔1B
You take one after-school dance class each week at a cost of £1/class leaving you 3 h on that evening for other leisure activities

You take two after-school dance classes each week at a cost of 50p/class leaving you 2 h on those evenings for other leisure activities

2A↔2B
You take two after-school dance classes each week at a cost of £1/class leaving you 2 h on those evenings for other leisure activities

You take one after-school dance class each week at a cost of 50p/class leaving you 3 h on that evening for other leisure activities

3A↔3B
You take one after-school dance class each week at a cost of 50p/class leaving you 2 h on that evening for other leisure activities

You take two after-school dance classes each week at a cost of £1/class leaving you 3 h on those evenings for other leisure activities

4A↔4B
You take two after-school dance classes each week at a cost of 50 p/class leaving you 3 h on those evenings for other leisure activities

You take one after-school dance class each week at a cost of £1/class leaving you 2 h on that evening for other leisure activities

Four paired choice scenarios were administered to 210 girls in seven schools—three intervention schools (n=90) and four control schools (n=120). Measures were taken at baseline (time 0) and at 9 weeks (time 1) using large cards, and girls' preferred choice for each pair of scenarios was recorded by the project team. Two time points were needed to establish change in preferences before and after the intervention. Participants were also asked to give preference ratings for 10 leisure activities on weekdays by survey using a 10-point scale (1=favourite; 10=least favourite). Participant responses were collected on Personal Digital Assistants (PDAs) and downloaded to a customised database.

## Resource use cost

At the start of this pilot study, there was minimal evidence on which to draw in identifying costs that might be included in a resource use checklist. One report from the National Institute for Health and Care Excellence (NICE) had modelled the cost of delivering dance classes to young children and produced some ball park cost estimates.[18][19] These were based on an account of the resources used in delivery of a dance programme for girls by Hampshire Dance and Trinity Laban[20] in which resources had been identified, but not costed. These uncontrolled studies provided a starting point and an opportunity to produce more complete and accurate costing data from the BGDP feasibility pilot trial in which the volume of resources used and prices could be treated separately. The cost items identified by NICE were entered in a database and data collected

using time sheets and expense sheets were collected by the project team. These cost estimates and some estimates for teacher time to manage behaviour derived by the first author of this article are detailed in table 2.[18–20] Table 2 was used as a template for identifying and costing resources in the BGDP feasibility study.

## Ethics

Potential participants in all seven schools were told that there was a maximum of 30 randomly assigned spaces at the dance classes. Informed parental consent was obtained for all participants.

## Analyses

Proportions of the sample ranking 10 weekday leisure activities as first choice activity (rank = 1) were calculated after participants had rated all 10 leisure activities from 1 to 10. Responses from participants with repetition of ratings for one or more leisure activities or missing ratings for leisure activities were excluded. Overall, where the proportion of the sample rating activities as their first choice was the same, these activities were assigned the same rank across all 10 activities. DCE data were 'effects-coded'[21] using STATA[22] and analysed using conditional logistic regression. Effects coding is similar to dummy variable coding, but is preferable in this instance because interaction or trade-off between the attributes is likely to take place as well as a main effect. The coefficients for each attribute are a measure of the influence of that attribute level on choice. Positive values represent a positive influence on choice, or in

**Table 2** Resources use identification template used to inform BGDP feasibility study

**NRG Youth Dance & Health Project***

| | †Total cost in £ |
|---|---|
| Project planning work: initial research into existing action research projects | 500.00 |
| Lead artist fee—programme design/artist training | 800.00 |
| Artists' travel fees—attending training/planning sessions | 637.35 |
| Artists' fee | 5806.00 |
| Artists' travel costs | 1515.44 |
| Coach hire—school group for pilot session | 562.88 |
| Space hire | 254.70 |
| Disclosures/refreshments | 77.98 |
| Postage | 64.03 |
| Management fee | 4000.00 |
| Staff travel | 443.65 |
| Documentation (dissemination advocacy) | 269.70 |
| Road-show event—end of project | 562.88 |
| Additional schools workshop | 151.80 |
| Total 2005–2006 prices in £ | 15 203.53 |
| Teacher time for behaviour management (not included in NRG report)‡ | 3300.00 |
| Total 2007–2008 prices in £ with teacher management | 19 427.76 |
| Total 2010–2011 prices in £ | 20 600.00 |

*Resource use items identified by Hampshire Dance and Trinity Laban, NRG Youth Dance and Health Project Evaluation report.[20]
†Assumptions and costing profile produced by Fordham and Barton[18] for NICE Guidance 17 (NICE, 2009).[19]
‡This item was identified in the NICE report[20] but not costed.[18] [19]
BGDP, Bristol Girls Dance Project; NICE, National Institute for Health and Care Excellence.

other words, a preference for that level of an attribute. These results can be used to establish girls' overall preferences for attributes, as well as the order of their preferences (ie, which attribute is most and least important). Participants with missing data were excluded from the DCE analysis.

Total and average cost estimates from a funder perspective were identified and derived for BGDP based on staged timing, quantity, frequency and price of resource use in 2010–2011 prices. Expenses including travel, intervention programme entry incentives, postage and Criminal Records Bureau (CRB) applications were accessed from the database maintained by the project team. Girls in the control schools received small thank you gifts at each data collection they attended. Space hire did not incur costs, but estimates of the cost of space hire for dance class delivery are included because they are costs connected with alternative use of space in schools. School overhead and capital costs are not included.

Grouping costs to enable estimation of the mainstream cost adopted the categories used in ASSIST (A Stop Smoking in Schools Trial).[23] Stage 0 intervention planning, development and training costs, stage 1 intervention preparation, stage 2 intervention delivery and stage 3 intervention maintenance costs were separately identified. Training costs for dance teachers are identified separately. Costs associated with running the research study, control group incentives for data collection, control school dance workshops and recruitment events would not recur during mainstream implementation, but these costs are included for clarity and completeness. All costs connected with tasks undertaken by the research team are not included.

## RESULTS
### Identification and timing of resources used
Table 3 identifies and describes at four stages the resources use of the BGDP programme and presents total cost estimates. The proportion of total costs incurred were 41% at stage 0, 7% at stage 1, 46% at stage 2 and 6% at stage 3. At stage 0, half of the costs are dance teacher preparation and training time which arguably would be incurred in part in delivery of a mainstream form of the programme. Eighteen BGDP dance classes (2 classes/week for 9 weeks) of 90 min duration were delivered to 90 girls in three intervention schools (30/ school) for 81 h (27 h/school) at a total estimated cost of $6380, £3988 and €4666 in 2010–2011 prices.[24] [25] The average cost of the BGDP programme in its mainstream form per school was $2126.40, £1329 and €1555 and per participant was $70.90, £44.31 and €51.84 in 2010–2011 prices. If training costs for dance teachers on the BGDP were included to the mainstream cost, this would add $1280, £800 and €928 to the cost per school and $43, £27 and €31.60 to the cost per pupil. These are not insubstantial additions, but are at the high end of training costs because this new dance programme was properly prepared for delivery. Training costs for the delivery of an established dance programme are likely to be lower. It was not possible to calculate CIs for average programme costs per school or per participant (see Limitations of this pilot study).

**Table 3** Description of resources used, unit volumes, prices and estimated costs* by category, by school and by pupil in 2010–2011 prices

| Stage of BGDP | Description of resources used | Number of units† | Price/unit‡ | Cost in £ 2010–2011 prices | Cost stage |
|---|---|---|---|---|---|
| Preprogramme planning development stage 0 | Lead dance artist consultation and development work | 2 days | £500/day | 1000 | 4726 |
| | Lead dance artist time, to adapt training programme for hip-hop genre | 1 day | £500/day | 500 | |
| | Lead dance artist time, to prepare dance teachers for 9 week intervention | 1 day | £500/day | 500 | |
| | 3 dance teachers preparation/training time | 7.5 days | £27/h | 2400 | |
| | Travel expenses | | | 288§ | |
| | Disclosures CRB forms | | | 38§ | |
| Programme preparation stage 1 | Space hire for dance taster sessions in intervention schools | 6 h | £15/h | 90 | 819 |
| | Dance teacher delivery of taster sessions in intervention schools | 6 h | £27/h | 162 | |
| | Control schools (n=4) recruitment presentations | 1 day | 500/day | 500 | |
| | Postage costs | | | 25§ | |
| | Travel expenses | | | 42§ | |
| Programme delivery stage 2 | Delivery 18 dance classes at 90 min/class over 9 weeks×3 schools | 81 h | £27/h | 2187 | 5375 |
| | T-shirts for 3 intervention schools | 90 girls | £5/T-shirt | 450 | |
| | Small gift incentives for control schools data collection | 3 waves | £731/wave | 2192 | |
| | Refreshments | | | 100§ | |
| | Travel expenses | | | 446§ | |
| Programme reinforcement stage 3 | Half-day dance workshops for two control schools | 9 h | | 243 | 729 |
| | 6 h performance events for parents of girls at 3 intervention schools | 18 h | | 486 | |
| | | | Total | | £11 649 |

| | |
|---|---|
| BGDP stages 0–3 costs in £s | £11649, 2010–2011 prices |
| BGDP stages 1–3 costs in £s | £6923, 2010–2011 prices |
| BGDP stages 1–3 minus control costs in £s | £3988, 2010–2011 prices |
| BGDP cost/school in £s | £1329, 2010–2011 prices |
| BGDP cost/pupil in £s | £44.31, 2010–2011 prices |

*Research team administration, travel and other costs are not included. Control costs in this research are included for information.
Sources:
†Research team.
‡Project budget—all prices are actual rates paid.
§Project budget—all costs are actual expenses incurred.
BGDP, Bristol Girls Dance Project; CRB, Criminal Records Bureau.

The shorter and less-intensive NRG programme costed in the NICE report[19] comprised 10 dance classes of 60 min duration for 24 girls (an assumed number) in 14 groups or schools (n=336 girls) for 140 h with a cost per participant of $98.14, £61.31 and €71.81 in 2010–2011 prices. This includes 140 h of teacher time sourced from national pay scales for England at £23.57/h at 2010–2011 prices.[26]

## Preferences for leisure activities

All girls were asked to rank 10 after-school leisure activities by first preference activity.

Table 3 presents proportions across the participants (n=210) for preference ranks for after-school leisure activities for all group allocations at each time point demonstrating consistency in preference ranks indicating girls' selection of first choice leisure activities at each time point. The after-school leisure activities indicating the highest proportion of first choice preference rankings at each time point include 'hanging out with friends away from home just for fun' (ranking at t2=1, t1=1 and t0=2); 'take part in sports, athletics or PA' (ranking at t2=2, t1=2 and t0=1) and 'using the Internet for fun: chats, YouTube, Facebook, Bebo, Myspace,

looking for music' (ranking at t0=3, t1=3 and t2=2). Valid responses were included in the analyses. Valid responses as a proportion of total responses for the survey ranking leisure activities were t2=178/210, t1=130/210 and t0=68/210 across all group allocations indicating particularly at baseline the participants experienced some problems using a hand-held PDA to rank and rate the weekday after-school leisure activities.

## DCE results

The p values for the regression coefficients in table 4 indicate that girls in this sample have a preference for 'time left for other leisure activities on dance class days', over the 'cost of' and 'frequency of dance classes per week'. Analysis of preference levels within each attribute suggests that 2 h is preferred to 3 h remaining for other leisure activities on dance class days. This pattern was consistent in all intervention and control groups at t0 and t1. Girls were least concerned with the frequency of dance classes per week with preference proportions suggesting two classes were preferred to one dance class per week in both intervention groups and the baseline control group (table 5).

## DISCUSSION

### What is already known on this topic

There is minimal guidance to support how economic evaluations of complex public health interventions should be designed and conducted in school and community settings.

There are no checklists or tools available to support costing dance programmes and minimal knowledge of how to categorise resources to identify the mainstream cost of delivery.

DCE methods to elicit the relative preferences and choices of girls aged 11–12 years are untried and

untested, but it is important to capture how girls value dance among other competing leisure activities using a robust and acceptable method.

### What this study adds

Providing programme cost data for a full trial of the BGDP programme is feasible, practical and likely to be successful. Around two-thirds of resources are development and research control costs, so resources used to develop, prepare and deliver these programmes should be categorised separately so that the cost of the mainstream programme can be estimated accurately.

DCE is an acceptable method to elicit preferences of girls aged 11–12 years.

At this point in their lives, after-school dance is an activity valued by girls when offered within the context of other competing choices and parental support for activities for spending leisure time after school on weekdays.

Participation in after-school dance classes has opportunity costs for participants and parents extending beyond the funder that suggest a social model of cost should be considered for to capture the costs associated with intervention outcome.

Robust evidence for the cost-effectiveness of PA complex interventions is important for knowing where to invest scarce resources and commission programmes to maximise health outcomes in primary prevention.[27–29] However, gathering robust evidence to support investment in public health interventions is a challenge.[30 31] Significant barriers remain and there is currently little guidance in how to conduct economic evaluation where behaviour change is associated with health outcomes determined beyond genetic inheritance by family, social and physical environments.[32 33]

Indicative programme cost data from the pilot economic evaluation indicated that a substantial proportion of the intervention programme costs, that is 41%, occurs

**Table 4** Preference rankings of first choice leisure activities at each time point N (%)

| After-school leisure activity | Time 2 | | Time 1 | | Baseline time 0 | |
|---|---|---|---|---|---|---|
| | Ranking | N (%) | Ranking | N (%) | Ranking | N (%) |
| Go around with friends to shopping centres, streets, parks just for fun | 1 | 46 (26) | 1 | 33 (25) | 2 | 12 (18) |
| Use the Internet for fun: chats, YouTube, Facebook, Bebo, Myspace, looking for music (do not include school homework) | 2 | 31 (17) | 3 | 20 (15) | 3 | 8 (12) |
| Take part in sports, athletics or physical activity | 2 | 31 (17) | 2 | 22 (17) | 1 | 13 (20) |
| Play with or see friends at your home or their homes | 3 | 21 (12) | 4 | 11 (9) | 5 | 5 (7) |
| Read books for enjoyment (do not include school books) | 4 | 13 (7) | 5 | 10 (8) | 4 | 6 (9) |
| Go to discos or dance classes | 5 | 11 (6) | 8 | 5 (4) | 5 | 5 (7) |
| Play a musical instrument, sing, draw, paint or write | 6 | 9 (5) | 4 | 11 (9) | 3 | 8 (12) |
| Send text messages or use Twitter on your mobile phone | 7 | 8 (5) | 7 | 6 (5) | 5 | 5 (7) |
| Play computer games | 8 | 4 (2) | 6 | 8 (6) | 6 | 4 (6) |
| Watch TV, DVDs or playbacks of programmes | 8 | 4 (2) | 9 | 4 (3) | 7 | 2 (3) |
| Total of valid* responses/total responses | | 178/210 | | 130/210 | | 68/210 |

*A valid response=each after-school leisure activity is ranked by a separate number between 1 and 10 by each individual participant using a PDA.
PDA, Personal Digital Assistant.

**Table 5** Regression coefficients indicating the value of dance classes at t0 (week 0) and t1 (week 9) by group allocation

| | Control time 0 (n=104/120*) | | | Control time 1 (n=104/120*) | | | Intervention time 0 (n=80/90*) | | | Intervention time 1 (n=80/90*) | | |
|---|---|---|---|---|---|---|---|---|---|---|---|---|
| | Coefficient | SE | p Value | Coefficient | SE | p Value | Coefficient | SE | p Value | Coefficient | SE | p Value |
| Frequency of dance class | | | <0.01 | | | <0.01 | | | <0.01 | | | 0.04 |
| Twice a week | **0.18** | 0.07 | | **0.30** | 0.07 | | **0.25** | 0.58 | | **0.13** | 0.06 | |
| Three times a week | −0.18 | 0.07 | | −0.30 | 0.07 | | −0.25 | 0.58 | | −0.13 | 0.06 | |
| Cost | | | <0.01 | | | <0.01 | | | <0.01 | | | <0.01 |
| £1 | **0.22** | 0.05 | | −0.17 | 0.04 | | **0.46** | 0.07 | | **0.26** | 0.06 | |
| £2 | −0.22 | 0.05 | | **0.17** | 0.04 | | −0.46 | 0.07 | | −0.26 | 0.06 | |
| Other hours available for leisure activities on dance class days | | | <0.01 | | | <0.01 | | | <0.01 | | | <0.01 |
| 2 h | **0.35** | 0.06 | | −0.31 | 0.65 | | **0.76** | 0.11 | | **0.37** | 0.08 | |
| 3 h | −0.35 | 0.06 | | **0.31** | 0.65 | | −0.76 | 0.11 | | −0.37 | 0.08 | |

Preferred level of attribute in bold.
*Number of valid responses from total possible responses.

at stage 0—the preprogramme development stage. This is an important finding because it suggests that the provided BGDP is effective and cost-effective in a full trial, and it would be substantially less costly to roll out in its mainstream form. All complex interventions in primary prevention are likely to generate a high proportion of upfront development costs that will not reoccur once a programme is mainstreamed—an aspect of investment in public health interventions often overlooked by decision-makers.

Application of DCE is an established technique in adult populations, but, to our knowledge, has not been applied previously in populations of children aged 11–12 years to establish values for the attributes of PAs. This study has demonstrated that application of DCE methods is feasible and acceptable to girls of this age. This is important because it suggests DCE could be applied in other studies with children to understand the concept of 'value' of an activity which plays an important role in recruitment, participation and maintenance of participants which are all linked to intervention outcome. In addition to its acceptability in this study, the DCE method has produced more complete and valid data than the direct survey method in eliciting preference ranks for after-school leisure activities. These findings support a previous contention that DCE techniques may have merit over more 'traditional' survey methods[34] in eliciting preferences. However, more evidence would be required to fully support this finding.

Taken together, findings of the DCE and survey of leisure activity preference in this study indicate that dance is a valued leisure activity among competing alternatives and reveals more about the attributes of dance classes in girls of this age that can be taken forward to maximise recruitment and retention in the BGDP programme. The findings of this study suggest that dance has immediate appeal as an after-school leisure activity among a range of strongly competing alternatives in girls of this age compared with older adolescents.[35] Girls in this study have a first rank preference for the attribute 'time remaining for other leisure activities on dance class days', over the 'cost of' and 'frequency of dance classes per week'. The finding that, in the intervention group, 2 h is preferred to 3 h remaining for other leisure activities on dance class days is significant. Overall, these findings could suggest that at this point in their lives dance is valued by girls as a physical and social activity when offered within the context of competing and constrained choices for spending leisure time at this age. For example, at this age, girls are not likely to be able to go to 'discos or dance classes' without parents or carers or to 'hang around on street corners with friends' and these issues may have affected their responses in the survey. These are important findings because they predict positive recruitment rates and participation of girls aged 11–12 years in dance as a physical leisure time activity and in a full trial.[36]

Delivery of after-school dance classes is dependent on substantial commitment from the girls giving up their after-school leisure time to participate in dancing. In turn, participation is dependent on the willingness of parents and carers to support attendance and to provide encouragement and a means of travelling back home after school hours when school buses are not available. This pilot study suggests development of a social model of costing that reflects the cost of participants' and parents' time and opportunity costs as substantial elements of the intervention cost that could be captured, if practical, in a full trial.

However, methods and tools to capture 'hidden' cost items that facilitate the success of the intervention, but are not incurred by funders, are not yet fully established.[23] Where to include training costs in these metrics is a question that remains for a future trial as they should arguably be included in mainstream cost estimation despite their categorisation as development costs. How identification of costs falling outside the public sector that are relevant to programme implementation can be captured at a full trial stage also needs to be considered carefully.[34] In a full trial, resources used should be captured prospectively[37] and this pilot study has established that categories of resource use are also important to consider to establish accurate mainstream programme costs.

## CONCLUSIONS

The feasibility of providing costing data for full trial of the BGDP programme is established and an embryonic resource use checklist has been developed. Resources used to develop and run the BGDP programme should be categorised separately in order for the mainstream delivery cost of BGDP to be estimated accurately in a full trial. A social model of costing that reflects participants' and parents' opportunity costs should be considered. BGDP after-school dance classes have potential for sustained participation and cost-effective delivery, but a full trial using methodological learning from this study is required.

**Acknowledgements** The authors would like to thank all the participants, dance teachers, school teachers and schools who participated in or assisted with this research. Professor JEP acknowledges discussion of and feedback from a draft version of this article presented at the Health Economists Bristol (HEB) journal club http://www.bris.ac.uk/social-community-medicine/centres/healthecon in November 2011. Laura Davis, project manager Bristol Girls Dance Project (BGDP), and Jade McNeil, research associate, are acknowledged for data collection leading to this article. FEC's contemporaneous PhD research was supervised by Dr Alan A Montgomery at the time of discrete choice experiment design and data analysis. This work was also supported by DECIPHer, a UKCRC Public Health Research: Centre of Excellence. Funding from the specific funders under the auspices of the UK Clinical Research Collaboration is gratefully acknowledged. The other coapplicants of the main BGDP study, Professor Ashley R Cooper and Dr Alan A Montgomery are acknowledged.

**Contributors** The quantitative analysis plan was conceived by JEP, FEC, RJ and SJS. Analyses were performed by JEP and FEC and interpretation by JEP, FEC, RJ, SJS and AMH. JEP drafted the first version of the manuscript with additional sections provided by FEC. All authors provided critical edits and revisions to the article and have reviewed and approved the final version of the article.

**Funding** This research was funded by the National Prevention Research Initiative (http://npri.org.uk), consisting of the following funding partners: Alzheimer's Research Trust; Alzheimer's Society; Biotechnology and Biological Sciences Research Council, British Heart Foundation; Cancer Research UK; Chief Scientist Office; Scottish Government Health Directorate; Department of Health; Diabetes UK; Economic and Social Research Council; Engineering and Physical Sciences Research Council; Health & Social Care Research & Development Office for Northern Ireland; Medical Research Council; the Stroke Association; Welsh Assembly Government and World Cancer Research Fund (grant number G090084).

**Competing interests** RJ is supported by a National Institute for Health Research career development fellowship.

**Ethics approval** This study was approved by a University of Bristol Ethics Committee.

**Provenance and peer review** Not commissioned; externally peer reviewed.

**Data sharing statement** No additional data are available.

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
