## [Reviewer comments · BMJ Open]

Some articles will have been accepted based in part or entirely on reviews undertaken for other BMJ Group journals. These will be reproduced where possible.

ARTICLE DETAILS

TITLE (PROVISIONAL)	Bristol Girls Dance Project feasibility study: using a pilot economic evaluation to inform design of a full trial
AUTHORS	Powell, Jane; Carroll, Fran; Sebire, Simon; Haase, Anne; Jago, Russ

VERSION 1 - REVIEW

REVIEWER	Jennifer R. O'Neill University of South Carolina, United States of America
REVIEW RETURNED	15-Sep-2013

GENERAL COMMENTS	Bristol Review 9-14-13 This article determined the economic costs of implementing a pilot dance intervention in 11-12 year old girls. This is an important research question due to limited information about the costs of implementing a physical activity intervention. The article is well-written. I have a few minor comments; please see below. Minor comments: 1. Abstract: the primary outcome is listed as MVPA minutes per day, however, this outcome is not reported in this paper. Although the purpose of the dance intervention was to increase MVPA minutes per day, the primary outcome of this paper seems to be the cost of implementing the dance program.2. Background, page 4, lines 16-17: "This suggested that it is feasible..." What is "this"? The meaning of the first part of this sentence is unclear.3. Background, page 4, line 36: Please define "DCE" here.4. Methods, page 6, line 47: Why were there 2 control arms? Why was this considered a three-arm study? What did the control school participants receive?5. Results, page 10, lines 38-42: How were the costs of dance classes estimated? Specifically, how were the salaries for the dance teachers determined, and how do those salaries compare to those of teachers in private dance schools?
---

REVIEWER	Noel Cameron Loughborough University, UK
REVIEW RETURNED	19-Sep-2013

GENERAL COMMENTS	1. The title and the focus of the article are not consistent. The article focus (page 5) includes use of DCE and a third focus which does not make sense in its current form. "To present learning from reflections..." is not a phrase that has clear meaning for me and I suspect for few others. It needs to be rewritten to clarify its meaning. 2. The "key messages" are about what has been done and not about what was found. They are not, therefore, messages that emanate from the study. 3. The English is a times confusing and needs to be carefully edited. On page 8, for instance, are "Expenses...gathered" or are they "assessed" or "estimated"? In the same paragraph the penultimate sentence does not make sense and appears to be missing a final phrase. In the next paragraph there is a semi-colon completely misplaced. 4. It would appear that one of the results of this intervention is to increase the preferences for the girls to "hang around in shopping malls" after school rather than participate in physical activity sessions which is an interesting if unwanted outcome I would have thought. 5. I think this study is interesting but the description needs cleaning up and clarifying and would benefit from dropping the third focus.
--

REVIEWER	Colin Green University of Exeter, UK
REVIEW RETURNED	22-Sep-2013

GENERAL COMMENTS	This paper points to the important, and often/mostly neglected, issue of dissemination of pilot trial findings that inform cost and cost-effectiveness methods for a future trial. I note the authors have published elsewhere what would be regarded as the 'primary' findings from their pilot trial. However, that paper did not include reference to the important aims of the pilot trial that covered pilot of methods to collect the appropriate data, in the appropriate way, to estimate resource use associated with (and therefore costs for) the delivery of the intervention, and the broader development of the framework for economic evaluation alongside a future definitive RCT. This is a common scenario, and the Authors in this paper seek to disseminate the economic evaluation elements of the pilot trial. My broader view is that it would be helpful to have all of the pilot results summarised together, to draw attention to the necessary combination of effectiveness and cost-effectiveness methods and data to inform health policy. However, I am only too aware of the difficulties doing this, so I do fully support the publication plan here, to publish separately on this occasion. This paper presents a number of important aspects of the research,
---

namely the resource use/cost aspect, and the consideration of the relative preferences of the potential participants in the dance classes, for disco or dance versus other activities, and for the particular characteristics (attributes) of the dance classes themselves. I do think that both of these aspects are interesting, and of importance in the development of a full RCT with CEA alongside. I must highlight that the current paper is not as clear (to read) as I feel it could be, in my current reading of it, and I feel its flow and readability could be improved with some further consideration from the Authors. The Reviewer checklist here refers to 'Research Question' and I don't see this clearly set out (I do see some insights in the strengths/limitations section, p6) – this would be a minor revision – and I would encourage Authors to consider that further (e.g. my view is that the RQ for this pilot trial is related to testing whether the proposed methods for collection of resource use data, for estimation of intervention costs, are feasible, practical, and likely to be successful in a future study, With the secondary topic of preferences in the context of the intervention set out a little clearer i.e. is this area of research informing the design of the intervention, or the 'relative value' of the intervention to the participants?). This aspect of the write-up overlaps a little with the 'study design' item in the Reviewer check-list, and the item on the clarity of methods, where I feel there is some potential for the Authors to be clearer on what they did to collect resource use data (currently refer to ASSIST, ref 17).

Where I would particularly like to see some additional information is on the presentation of results (statistics?) for both the cost estimates and the DCE results. I appreciate there is a challenge in covering all areas in detail, within word limits. However, I feel that presentation of results for costing would be more transparent if the Table/Results allowed the reader to see the breakdown of the cost estimates for intervention (what needs to be done if intervention introduced) and control (which could be interpreted as research related). For example, Table II, £4,789 sub-total consists of ? (stage 2 research costs are £5,982, how does the reader know what is going on there?). I would suggest providing the reader with unit costs used, and itemising resource use, for intervention particularly, to show how the cost estimate breaks down.

Another related question is on the training aspects, ... would this be needed for roll out of the intervention? Is some aspect of cost needed within the intervention estimate to capture that? (although I accept this would make only a very small change to the mean cost per person).

I feel the ranking data is not/may not be clearly interpreted, and could benefit from further discussion, or discussion of the relative merits of the ranking data. At the moment the results are in Tabular form, and the text (p11, 34-55) does not draw attention to (other than implicitly) the fact that dance (disco or dance) is ranked 5 or below, and consistently below sports/other physical activity. The discussion (p13, 13-18) refers to dance as a "popular choice", but some additional support may be needed for this judgement.

I feel it would be helpful to see the regression output/coefficients for the DCE data/study, as this is the typical way that such studies are presented.

The preference study is very interesting. It could be a more detailed individual publication, but I do find it helpful to see presentation in the context of the broader aims of the pilot trial, and find the message over the acceptability of DCE methods to this age group

	(participant group) an important message. In summary I think the Authors may (should) be able to present a clearer paper that is of interest and value in communicating the learning from the pilot study to inform a future economic evaluation, and to seek to highlight that such findings could/should be published more commonly (although, in my view ideally as part of the main/primary findings for the pilot study).
--	--

VERSION 1 – AUTHOR RESPONSE

Reviewer 1

This article determined the economic costs of implementing a pilot dance intervention in 11-12 year old girls. This is an important research question due to limited information about the costs of implementing a physical activity intervention. The article is well-written.

Response: Thank you

I have a few minor comments; please see below.

1. Abstract: the primary outcome is listed as MVPA minutes per day, however, this outcome is not reported in this paper. Although the purpose of the dance intervention was to increase MVPA minutes per day, the primary outcome of this paper seems to be the cost of implementing the dance program.
Response: This sentence has been removed from the abstract and the economic outcome measures clarified.

2. Background, page 4, lines 16-17: “This suggested that it is feasible...” What is “this”? The meaning of the first part of this sentence is unclear.
Response: The text has been edited to make it clear that some of the findings of this feasibility study have been reported elsewhere.

3. Background, page 4, line 36: Please define “DCE” here.
Response: This term has been defined at first point of mention in the text.

4. Methods, page 6, line 47: Why were there 2 control arms? Why was this considered a three-arm study? What did the control school participants receive?
Response: Explanation for having 2 control arms has been added to the text with details of the incentives for data collection in each control arm given.

5. Results, page 10, lines 38-42: How were the costs of dance classes estimated? Specifically, how were the salaries for the dance teachers determined, and how do those salaries compare to those of teachers in private dance schools?
Response: A new table has been added to show the origin of the resource use checklist for the cost items in the BGDG feasibility study (Table II). Table III (Table II in first version) has been revised to show how the costs of the dance classes have been estimated and break down. Actual costs incurred or rates paid for dance teachers and lead artist consultancy has been made clear in Table III. Research costs connected with the research team and tasks they completed have been removed for clarity and control costs and intervention costs have been identified separately.

Reviewer 2

1. The title and the focus of the article are not consistent. The article focus (page 5) includes use of DCE and a third focus which does not make sense in its current form. “To present learning from reflections...” is not a phrase that has clear meaning for me and I suspect for few others. It needs to be rewritten to clarify its meaning.

Response: The focus of the article has been amended to match the title of the paper. The third focus has been removed.

2. The “key messages” are about what has been done and not about what was found. They are not, therefore, messages that emanate from the study.

Response: This section has been amended to reflect specific rather than more general findings from this pilot study.

3. The English is a times confusing and needs to be carefully edited. On page 8, for instance, are “Expenses...gathered” or are they “assessed” or “estimated”? In the same paragraph the penultimate sentence does not make sense and appears to be missing a final phrase. In the next paragraph there is a semi-colon completely misplaced.

Response: these sections have been completely rewritten and re-ordered. The semi-colon has been replaced by a comma.

4. It would appear that one of the results of this intervention is to increase the preferences for the girls to “hang around in shopping malls” after school rather than participate in physical activity sessions which is an interesting if unwanted outcome I would have thought.

Response: The latter part of the discussion section has been amended to contextualise this possible interpretation. The validity of the survey data has been discussed more fully in the results section under the sub-heading ‘preferences for leisure activities’.

5. I think this study is interesting but the description needs cleaning up and clarifying and would benefit from dropping the third focus.

Response: Thank you. We have dropped the third focus and improved the description of the study.

Reviewer 3

This paper points to the important, and often/mostly neglected, issue of dissemination of pilot trial findings that inform cost and cost-effectiveness methods for a future trial. I note the authors have published elsewhere what would be regarded as the 'primary' findings from their pilot trial. However, that paper did not include reference to the important aims of the pilot trial that covered pilot of methods to collect the appropriate data, in the appropriate way, to estimate resource use associated with (and therefore costs for) the delivery of the intervention, and the broader development of the framework for economic evaluation alongside a future definitive RCT.

This is a common scenario, and the Authors in this paper seek to disseminate the economic evaluation elements of the pilot trial. My broader view is that it would be helpful to have all of the pilot results summarised together, to draw attention to the necessary combination of effectiveness and cost-effectiveness methods and data to inform health policy. However, I am only too aware of the difficulties doing this, so I do fully support the publication plan here, to publish separately on this occasion.

Response: Thank you for drawing attention to the difficulties often encountered by economists in contributing to trials and supporting a separate publication on this occasion.

This paper presents a number of important aspects of the research, namely the resource use/cost aspect, and the consideration of the relative preferences of the potential participants in the dance classes, for disco or dance versus other activities, and for the particular characteristics (attributes) of the dance classes themselves. I do think that both of these aspects are interesting, and of importance in the development of a full RCT with CEA alongside.

Response: Thank you, we agree.

I must highlight that the current paper is not as clear (to read) as I feel it could be, in my current reading of it, and I feel its flow and readability could be improved with some further consideration from the Authors. The Reviewer checklist here refers to ‘Research Question’ and I don’t see this clearly set out (I do see some insights in the strengths/limitations section, p6) – this would be a minor revision – and I would encourage Authors to consider that further (e.g. my view is that the RQ for this pilot trial is related to testing whether the proposed methods for collection of resource use data, for estimation of intervention costs, are feasible, practical, and likely to be successful in a future study, With the secondary topic of preferences in the context of the intervention set out a little clearer i.e. is this area

of research informing the design of the intervention, or the 'relative value' of the intervention to the participants?).

Response: Thank you for these very helpful observations which have helped us to clarify, structure and edit our revised paper. The research questions are set out under 'article focus' in the 'article summary'.

This aspect of the write-up overlaps a little with the 'study design' item in the Reviewer check-list, and the item on the clarity of methods, where I feel there is some potential for the Authors to be clearer on what they did to collect resource use data (currently refer to ASSIST, ref 17).

Response: Thank you for pointing out this issue which has led to a substantial improvement to this paper and resulted in a complete rethink about how to present these data. The context for, process of and origin source for identifying and describing the resources to deliver the BGDP have been added to the text. A new table has been added Table II. Having given these explanations and additional data - it is hoped that the methods used for estimating costs can be understood more clearly. Table II has been amended (now Table III) to clarify how costs have been estimated for intervention and control schools and expenses data used. Sources for prices have been given. Research costs of the research project team have been removed to aid clarity and those costs relating to control aspects are separately identified and estimated.

Where I would particularly like to see some additional information is on the presentation of results (statistics?) for both the cost estimates and the DCE results. I appreciate there is a challenge in covering all areas in detail, within word limits. However, I feel that presentation of results for costing would be more transparent if the Table/Results allowed the reader to see the breakdown of the cost estimates for intervention (what needs to be done if intervention introduced) and control (which could be interpreted as research related). For example, Table II, £4,789 sub-total consists of ? (stage 2 research costs are £5,982, how does the reader know what is going on there?). I would suggest providing the reader with unit costs used, and itemising resource use, for intervention particularly, to show how the cost estimate breaks down.

Response: Completed - see Tables III for costs and cost categories and V for the DCE findings.

Another related question is on the training aspects, ... would this be needed for roll out of the intervention? Is some aspect of cost needed within the intervention estimate to capture that? (although I accept this would make only a very small change to the mean cost per person).

Response: This is an interesting and pertinent question and does make some difference to the cost estimates. We demonstrate in the results section how the addition of these costs would alter the mean cost per school and per pupil and provide explanation. We mention this issue in the revised methods section and add it as a question remaining to the discussion section.

I feel the ranking data is not/may not be clearly interpreted, and could benefit from further discussion, or discussion of the relative merits of the ranking data. At the moment the results are in Tabular form, and the text (p11, 34-55) does not draw attention to (other than implicitly) the fact that dance (disco or dance) is ranked 5 or below, and consistently below sports/other physical activity. The discussion (p13, 13-18) refers to dance as a "popular choice", but some additional support may be needed for this judgement.

Response: We have discussed the validity of the data in Table IV and contextualised / extended the discussion.

I feel it would be helpful to see the regression output/coefficients for the DCE data/study, as this is the typical way that such studies are presented.

Response: Table V (old table IV) has been replaced to reflect the regression output coefficients and p values.

The preference study is very interesting. It could be a more detailed individual publication, but I do find it helpful to see presentation in the context of the broader aims of the pilot trial, and find the message over the acceptability of DCE methods to this age group (participant group) an important message.

Response: Thank you, we agree with this comment and have amended the text throughout to highlight the importance of DCE as a useful and appropriate method at the feasibility pilot phase to inform the design of interventions.

In summary, I think the Authors may (should) be able to present a clearer paper that is of interest and value in communicating the learning from the pilot study to inform a future economic evaluation, and to seek to highlight that such findings could/should be published more commonly (although, in my view ideally as part of the main/primary findings for the pilot study).

Response: We agree. Thank you for showing understanding of the reasons for not doing so on this occasion.

VERSION 2 – REVIEW

REVIEWER	Colin Green University of Exeter, UK
REVIEW RETURNED	10-Nov-2013

GENERAL COMMENTS	The Authors have responded to Reviewers comments in a thorough and positive manner, and the paper is clearer in its structure and presentation. Minor note: Table II needs to be considered for alignment/format (the version I have is not aligned by row).
--